# MOF-Derived Ultrathin NiCo-S Nanosheet Hybrid Array Electrodes Prepared on Nickel Foam for High-Performance Supercapacitors

**DOI:** 10.3390/nano13071229

**Published:** 2023-03-30

**Authors:** Jing Li, Jun Li, Meng Shao, Yanan Yan, Ruoliu Li

**Affiliations:** School of Materials Science and Engineering, Shanghai University of Engineering Science, Shanghai 201620, China

**Keywords:** MOF, self-sacrificing template, transition metal sulfides, supercapacitor

## Abstract

At present, binary bimetallic sulfides are widely studied in supercapacitors due to their high conductivity and excellent specific capacitance (SC). In this article, NiCo-S nanostructured hybrid electrode materials were prepared on nickel foam (NF) by using a binary metal–organic skeleton as the sacrificial template via a two-step hydrothermal method. Comparative analysis was carried out with Ni-S and Co-S in situ on NF to verify the excellent electrochemical performance of bimetallic sulfide as an electrode material for supercapacitors. NiCo-S/NF exhibited an SC of 2081 F∙g^−1^ at 1 A∙g^−1^, significantly superior to Ni-S/NF (1520.8 F∙g^−1^ at 1 A∙g^−1^) and Co-S/NF (1427 F∙g^−1^ at 1 A∙g^−1^). In addition, the material demonstrated better rate performance and cycle stability, with a specific capacity retention rate of 58% at 10 A∙g^−1^ than at 1 A∙g^−1^, and 75.7% of capacity was retained after 5000 cycles. The hybrid supercapacitor assembled by NiCo-S//AC exhibited a high energy density of 25.58 Wh∙kg^−1^ at a power density of 400 W∙kg^−1^.

## 1. Introduction

Supercapacitors with large charge storage capacity and good long-term cyclical stability have aroused widespread concern to meet the needs of the power supply system due to the rapid development of portable electronic equipment [1,2,3]. The electrode materials, as the key component of supercapacitors, produce an essential effect on their service performance. Transition metal compounds (oxides, hydroxides, and sulfides) as promising electrode materials have attracted extensive attention owing to their low manufacturing cost, simple synthesis process, high theoretical specific capacity, and high energy density [4,5,6,7]. Moreover, they could provide enough storage charge during their contact reactions with the electrolyte owing to their varieties of chemical valence states [8]. Sulfur exhibits lower electronegativity than oxygen, making the transition metal sulfide carry lower band gap energy [9]. Thus, replacing oxygen with sulfur in transition metal compounds could effectively promote electron migration, resulting in the transition metal sulfide as the electrode material demonstrating more excellent electrical conductivity, better electrochemical activity, and higher long-term cycling stability than oxides and hydroxides. Furthermore, according to recent research reports [10,11], bimetallic sulfide presents better conductivity and exposes more electrochemical active sites than monometallic sulfide due to the synergistic effect between metal cations. Among all sulfides (such as NiCoS [12], ZnCoS [13], CuCoS [14], and CoMoS [15]), NiCoS with a representative AB_2_S_4_ structure has received the most attention [16]. In NiCo_2_S_4_, Ni^2+^ and Ni^3+^ occupy octahedral positions, and Co^2+^ and Co^3+^ occupy tetrahedral and octahedral positions, resulting in the formation of a typical spinel-structured binary metal sulfide [17]. NiCo_2_S_4_ has abundant mixed valence redox reactions (Ni^2+^/Ni^3+^ and Co^2+^/Co^3+^) and extremely high conductivity (1.23 × 10^−6^ Sm^−1^) [18,19]. NiCo_2_S_4_ with different nanostructures, such as 1D nanowires [20], 2D nanosheets [21], and 3D nanoflowers [22], have been synthesized and demonstrated unique electrochemical performance. Chen et al. [23] prepared NiCo_2_S_4_ nanosheets@nanowire on nickel foam (NF) by using a chemical liquid deposition method. The electrode had a specific capacity of 1777 F∙g^−1^, and its capacity retention rate was 83% after 3000 cycles with sweeping speed of 10 A∙g^−1^. SrinivasaRao [12] synthesized NiCoS nanospheres on NF in situ via a two-step hydrothermal method, and the findings demonstrated a specific capacitance (SC) of 922.06 F∙g^−1^ at 0.62 F∙g^−1^ and a capacity retention rate of 92.1% after 3000 cycles at 2.5 A∙g^−1^. Li et al. [24] successfully prepared NiCo_2_S_4_ with a hollow nanocube structure via a simple high-temperature reflux process (hydroxylation and subsequent vulcanization) and coated it on NF in the form of slurry to prepare electrode materials, which maintained a specific capacity of 1350 F∙g^−1^ at 1 A∙g^−1^. The assembled two-electrode device had a good cycle stability (67% retention rate after 10,000 cycles) and represented a high energy density (33 Wh∙kg^−1^ at 800 W∙kg^−1^).

For a given electrode material, its electrochemical performance is closely correlated with its specific surface area [25,26]. A high specific surface area could provide more active sites for the electrochemical reactions, contributing to the improvement in specific capacity. The charge migration distance is also greatly shortened, which is beneficial to the rating capability. Therefore, increasing the specific surface area of bimetallic sulfide by adjusting its morphology for enhanced electrochemical performance is essential. Metal–organic framework (MOF) is a porous crystal structure formed by the orderly connection of metal central sites and organic ligands, and it presents the characteristics of high specific surface area [27,28]. Transition metal compounds derived from MOF as a sacrificed template could inherit the excellent properties of MOF. Up to now, bimetallic sulfide (NiCoS) prepared by introducing MOFs as electrode materials for supercapacitors has attracted wide attention. Liu et al. [29] used Co-MOF as a template, prepared NiCo-LDH/CC via subsequent ion exchange method, and finally obtained nanosheet NiCoS/CC electrode materials via solvothermal vulcanization method. The electrode material exhibited an excellent specific capacity of 1653 F∙g^−1^ at 1 A∙g^−1^. In addition, the electrode material showed excellent rate performance (1275 F∙g^−1^ at 20 A∙g^−1^) and maintained 84% of the original specific capacity after 3000 cycles at 10 A∙g^−1^. Chen et al. [29] mainly used Ni-MOF microspheres as a template to prepare nanosheet-assembled flower-like Ni/Co-MOF by etching in cobalt nitrate solution and then prepared uniform Ni-Co-S nanoparticles by a further solvent-thermal vulcanization method. The samples were coated on NF in the form of slurry to obtain electrode materials. These electrode materials showed a higher SC of 1377.5 F∙g^−1^ at 1 A∙g^−1^ and better cycle stability (a retention rate of 93.7% after 3000 cycles) than Ni/Co-MOF (1220.2 F∙g^−1^ at 1 A∙g^−1^, a retention rate of 87.8% after 3000 cycles). Pang et al. [30] separately studied the electrochemical performance of a three-electrode system assembled by NiCo-S. NiCo-MOF-74 powder was prepared by controlling the molar ratio of Ni salt and Co salt via the hydrothermal method. The sample was annealed and oxidized for 2 h at 350 °C and then vulcanized via the hydrothermal method, finally obtaining rod-like NiCo-S with excellent electrochemical performance (specific capacity of 1910 F∙g^−1^ at 1 A∙g^−1^ and retention rate of 73.7% at 10 A∙g^−1^). The above research reports showed that MOF as a sacrificial template is a relatively excellent method to improve the electrochemical performance. However, according to recent studies, bimetallic sulfides derived from MOFs are mainly prepared into electrode materials by using traditional slurry coating processes. The structure of the prepared electrode materials may be destroyed to a certain extent during the grinding and tableting treatment [31]. In addition, the introduction of conductive agents and binders could increase the internal resistance of electrode materials [32,33], which could hinder charge migration and reduce the electrochemical performance. Electrode materials prepared by an in situ synthesis method could avoid the above disadvantages. Some investigations have been performed to in situ synthesize a single-metal MOF (Ni or Co) as a sacrifice template, followed by ion exchange or etching to produce the intermediate product and then hydrothermal vulcanization to produce bimetallic sulfide. The process is comparatively complicated, resulting in the structure and performance of resultant products being difficult to control.

In this study, the NiCo-MOF as a self-sacrificing template was first in situ synthesized on NF and then transformed into NiCo-S by the hydrothermal method to address the issues mentioned above. In the first step of the hydrothermal process, the NiCo-MOF with uniform distribution of metal ions was grown in situ on NF, and then a sulfur source was introduced to participate in the ion exchange reaction, which was maintained at 140 °C for 6 h to completely transform NiCo-MOF/NF into NiCo-S/NF in the second step of the hydrothermal process. On this basis, by adjusting the amount of Ni salt and Co salt, nanosheet@dendritic-like Ni-S/NF and nanosheet-like Co-S/NF were successfully synthesized. The change in the morphology and structure from Ni-S/NF and Co-S/NF to NiCo-S/NF was studied, and a relationship between this change and the electrochemical performance was established. The test and analysis of the three- and two-electrode systems obviously showed that the as-prepared nanosheet@nanoleave-like NiCo-S/NF electrode material exhibited excellent electrochemical performance.

## 2. Experimental

### 2.1. Materials

All chemicals of analytical grade were directly used without further purification. Potassium hydroxide (KOH), thiourea (CH_4_N_2_S), and ethanol (C_2_H_6_O) were supplied by Shanghai Titan Scientific Co., Ltd. (Shanghai, China). Cobalt nitrate hexahydrate (Co(NO_3_)_2_·6H_2_O), nickel nitrate hexahydrate (Ni(NO_3_)_2_·6H_2_O), 1,4-terephthalic acid (PTA), and N,N-dimethylformamide (DMF) were provided by Aladdin Industrial Corporation (Shanghai, China). Active carbon (AC) was obtained from Jiangsu Xianfeng Nano Material Technology Co., Ltd. (Nanjing, China). NF was obtained from Kunshan Guangjiayuan New Materials Co., Ltd. (Jiangsu, China).

### 2.2. Synthesis of NiCo-S on NF (NiCo-S/NF)

A simple two-step hydrothermal method was used to synthesize NiCo-S derived from MOF on Ni foam, and the procedures are illustrated in Figure 1. The NF (1.5 cm × 2 cm, thickness of 1.0 mm, porosity of 96.5%, and specific surface area of 320 m^2^·g^−1^) was progressively cleaned in an ultrasonic cleaner with acetone, 1 M HCl solution, deionized water, and absolute ethanol for 10 min to remove the surface oil and oxide layer. It was then dried in a vacuum oven at 60 °C. In the first step, the MOF as the precursor was prepared in situ via a facile hydrothermal method. Typically, Ni(NO_3_)_2_·6H_2_O and Co(NO_3_)_2_·6H_2_O with a mass ratio of 1:1 were dissolved in 20 mL DMF solution under stirring to form a homogeneous solution and then added into a mixed solution containing 10 mL DMF with 1 mmol PTA under magnetic stirring. The mixture was subsequently transferred into a 100 mL Teflon stainless autoclave with a piece of pretreated NF for the next hydrothermal treatment. The autoclave was maintained at 170 °C for 12 h. Subsequently, the produced NF was cleaned with DMF, deionized water, and absolute ethanol multiple times to remove the impurities and then dried for 12 h in a vacuum oven at 60 °C. The obtained product was named as NiCo-MOF/NF. The loading amount of the active materials in the working electrode was approximately 0.67 mg·cm^−2^.

In the second step, 0.1 g CH_4_N_2_S was dissolved in a 30 mL mixed solution of deionized water and absolute ethanol with a volume ratio of 1:1 and then transferred into a 100 mL Teflon stainless autoclave. The NiCo-MOF/NF sample was placed in the autoclave and maintained at 140 °C for 6 h. Once room temperature was reached gradually, the product was cleaned with deionized water and absolute ethanol several times and dried in a vacuum oven at 60 °C overnight. The obtained product was named as NiCo-S/NF. The loading amount of the active materials in the working electrode was approximately 1.08 mg·cm^−2^. For comparison, the same steps mentioned above were repeated to synthesize Ni-S/NF (without Co salt) and Co-S/NF (without Ni salt).

### 2.3. Material Characterizations

The crystal structure of the samples was characterized using X-ray diffraction (XRD, D2-PHASER, Bruker, Karlsruhe, Germany) with Cu-Kα radiation (γ = 0.1540560 nm). The elemental compositions and surface chemical states of the samples were analyzed by X-ray photoelectron spectroscopy (XPS, Thermo Fisher Scientific, Waltham, MA, USA) using Al-Kα X-ray as the excitation source. Scanning electron microscopy (SEM, Sigma 300, Zeiss, Jena, Germany) with energy dispersive spectroscopy (EDS, Xplore 30, Oxford, UK) and transmission electron microscopy (TEM, JEM-2100F, JOEL, Tokyo, Japan) were used to observe the morphologies of the samples.

### 2.4. Electrochemical Performance Tests

#### 2.4.1. Three-Electrode System

On the basis of the typical three-electrode configuration, the prepared samples of MOF-derived Ni-S, Co-S, and NiCo-S grown in situ on NF with a size of 1.5 cm × 2 cm were directly used as the working electrodes, while the graphite sheet (20 mm × 25 mm) and saturated calomel electrode were applied as the counter electrode and reference electrode, respectively. An aqueous solution of 6 M KOH was used as the electrolyte. The electrochemical performance of the electrode materials was tested by cyclic voltammetry (CV) on a CS350H electrochemical workstation (CS350H, CORRTEST, Wuhan, China), galvanostatic charge/discharge (GCD) on a CHI 760E electrochemical workstation (CHI 760E, CH Instrument Inc, Shanghai, China), and electrochemical impedance spectroscopy (EIS) on a CS350H electrochemical workstation (CS350H, CORRTEST, Wuhan, China). The CV measurements were performed at various scan rates between 5 and 50 mV·s^−1^ in the potential range from −0.2 to 0.5 V. The GCD tests were carried out at the current densities of 1, 2, 4, 6, 8, and 10 A·g^−1^ within a potential range of 0–0.37 V. The CV tests were conducted to measure the cycle stability of the electrode (NiCo-S derived from NiCo-MOF loaded with NF sheet) for 5000 cycles at 50 mV·s^−1^. The EIS tests were conducted from 0.01 Hz to 100 kHz with an amplitude of 5 mV at the open-circuit potential, and the resultant spectra were simulated using Zview software.

Then, on the basis of the GCD data, the SC ($C_{S}$, F∙g^−1^) of the electrode materials was calculated by the following formulas [7,10]:

For the GCD tests
(1)$$C_{S}=\frac{I\times △t}{m\times △V}$$
where m represents the mass of the active material (g), △V denotes the potential range (V), I is the discharge current (A), and △t is the discharge time (s).

#### 2.4.2. Two-Electrode System

The hybrid supercapacitor (HSC) device was assembled to detect the electrochemical performance of the electrode materials. In the HSC device, the NiCo-MOF/NF-derived NiCo-S/NF electrode material was used as the positive electrode, and the AC/NF electrode served as the negative electrode owing to its high specific surface and excellent electrical conductivity. A filter paper was used as a diaphragm in the middle of the positive and negative electrode to effectively prevent internal short circuit caused by contact between the two electrodes. Moreover, 6 M KOH was employed as an electrolyte. For the preparation of the negative electrode, N-methyl-2-pyrrolidone (NMP) solvent was mixed with activated carbon (90 wt.%) and polyvinylidene fluoride (10 wt.%) and then coated on an NF. The coated portion was then folded and pressed for 10 s at a pressure of 10 MPa. Finally, the prepared AC electrode was dried at 60 °C for 12 h and drenched in 6 M KOH solution overnight for electrochemical tests. Meanwhile, to maintain the conservation of positive and negative charges in the HSC device for enhancement of the electrochemical performance, the mass ratio of the positive and negative electrodes was balanced using the following formula [7]:(2)$$\frac{m^{+}}{m^{-}}=\frac{{C_{s}}^{-}\times {△V}^{-}}{{C_{s}}^{+}\times {△V}^{+}}$$
where $m^{-}$ (g), ${△V}^{-}$ (V), and ${C_{s}}^{-}$(F∙g^−1^) represent the mass, potential range, and SC of the negative electrode, respectively; $m^{+}$ (g), ${△V}^{+}$ (V), and ${C_{s}}^{+}$ (F∙g^−1^) denote the mass, potential range, and SC of the positive electrode, respectively. In light of the charge conservation in the formula, the mass ratio of positive and negative materials was 1:1.63.

For the two-electrode system, the same electrochemical workstation applied in the three-electrode system was used to test the electrochemical performance (CV, GCD, and EIS). The specific capacity ($C_{device}$, F∙g^−1^) of the HSC device was calculated by the following formula [26]:(3)$$C=\frac{\int I\left(V\right)dV}{2m\upsilon (△V)}$$
(4)$$C_{device}=\frac{I\times △t}{m_{total}\times △V}$$where I represents the discharge instantaneous current (A), υ denotes the scanning rate (mV·s^−1^), m represents negative active material (g), $m_{total}$ represents the total mass of positive and negative active material (g), △V denotes the potential range (V), and △t refers to the discharge time (s).

Energy density (E, Wh∙kg^−1^) and power density (P, W∙kg^−1^) are two crucial performance indicators to assess the practical application of the HSC device. The values of these two indicators were calculated by the following formula [17,34]:(5)$$E=\frac{C_{device}{\times △V}^{2}}{2\times 3.6}$$
(6)$$P=\frac{3600\times E}{△t}$$
where $C_{device}$ (F∙g^−1^) represents the specific capacity of the HSC device based on the GCD curves of the two-electrode system, and △V (V) and △t (s) are the potential range and discharge time, respectively.

## 3. Results and Discussions

### 3.1. Structural and Morphology Characterization

The powder products of Ni-S, Co-S, and NiCo-S growing on NF were dispersed into ethanol solution by ultrasound to eliminate the influence of NF on the XRD results of the samples. The sample powders were then tested and evaluated by XRD after vacuum at 60 °C overnight. The diffraction peaks are displayed in Figure 2. A comparison shows that the distinctive diffraction peaks of Ni-S, Co-S, and NiCo-S corresponded to Ni_3_S_2_ (JCPDS: 44-1418), Co_3_S_4_ (JCPDS: 42-1448), and NiCo_2_S_4_ (JCPDS: 73-1704), respectively, indicating that these are the crystalline phases of the prepared Ni-S, Co-S, and NiCo-S materials. The peaks at 21.8°, 31.1°, 37.8°, 44.4°, 49.7°, and 54.6° were defined as the (101), (110), (003), (202), (113), and (104) crystal planes of Ni_3_S_2_, respectively. The peaks at 31.1°, 37.8°, 49.9°, and 55.1° were the characteristic peaks of Co_3_S_4_, and they could be attributed to its (311), (400), (511), and (440) crystal planes, respectively. For NiCo_2_S_4_, diffraction peaks could be seen at 16.1°, 31.3°, 37.9°, 50.2°, and 55.1°, which could be defined as the characteristic peaks of (111), (311), (400), (511), and (440) crystal planes, respectively. NiCo-S had a lower diffraction peak intensity than Ni-S and Co-S under the same reaction circumstances, suggesting that the material was in a less crystalline form [35]. The peak of the three samples at 44.3° corresponded to Ni (JCPDS: 89-7128), which may be the mixed NF skeleton in the process of ultrasonic dispersion. Meanwhile, the Co-S sample without a Ni source could also produce a small amount of Ni_3_S_2_ (21.7°, JCPDS: 44-1418). The Ni_3_S_2_ core in this instance was assumed to have originated from the NF. Some vulcanization reactions took place in the foamed nickel of all samples during the vulcanization reaction [17]. Hong et al. [36] observed the characteristic peaks of NiCo-MOF (11.1°, 15.3°, and 23.6°), which was not shown in the XRD spectrum of the present study, indicating that all NiCo-MOF was transformed into NiCo-S. Overall, the NiCo-S/NF electrode material was successfully synthesized in this experiment.

XPS measurements were performed to provide further insights into the chemical compositions and chemical valence states of the prepared samples. The XPS survey spectrum of NiCo-S/NF is shown in Figure 3a. Ni 2p, Co 2p, S 2p, and O 1s C 1s could be observed in the spectrum, indicating the presence of these elements in the sample. The C element could be seen in the spectrum of NiCo-S/NF due to the influence of the test environment. Meanwhile, the appearance of O 1s peak was affected by the use of organic ligands in the experiment. As shown in Figure 3b, Ni 2p contained two pairs of spin orbitals (Ni 2p_1/2_ and Ni 2p_3/2_) and two satellite peaks (“Sat.”). The binding energy at 860.09 eV in Ni 2p_3/2_ and 873.78 eV in Ni 2p_1/2_ could be ascribed to the Ni^3+^ states, while those located at 852.54 and 870.77 eV could be assigned to the Ni^2+^ states [17,37]. Meanwhile, the accompanied weak peaks at 861.72 and 879.62 eV belonged to two satellite peaks (“Sat.”). These findings proved that the Ni atoms in NiCo-S had two different valence states, indicating the existence of Ni^3+^ and Ni^2+^. Similarly, in Figure 3c, the diffraction peaks at 781.6 and 797.55 eV could be assigned to Co^2+^, and the binding energies of the two satellite peaks were located at 787.88 and 803.38 eV. The peak values of Co^3+^ were located at 791.58 and 777.62 eV. The two pairs of diffraction peaks and satellite peaks indicated two different valence states of Co atoms in NiCo-S, Co^2+^, and Co^3+^, respectively [38]. In Figure 3d, the peaks at 163.8 and 162.27 eV in the S 2p spectra could be attributed to S 2p_1/2_ and S 2p_3/2_, respectively, indicating that the S atom in the sample material was S^2−^ [28]. Meanwhile, another peak at 168.65 eV could be assigned to the high-valence oxidation peaks of surface sulfur pieces. As described above, the XPS results further confirmed that NiCo-S consisted of Ni, Co, S, and O elements, and multiple valence states existed in the elements (Ni^2+^, Ni^3+^, Co^2+^, Co^3+^, and S^2−^), consistent with the related literature on NiCo-S/NF electrodes [35,39].

Figure 4 shows the microstructural characteristics of Ni-S/NF, Co-S/NF, and NiCo-S/NF, as analyzed by SEM. For Ni-S/NF, dendritic and flaky nanostructures were formed on the surface of NF (Figure 4a–c), but the growth direction was different, and the density was uneven. For Co-S/NF, the nanosheets were grown vertically on NF, showing a uniformly dense array of nanosheets with distinct sheets (Figure 4d–f). When the Ni^2+^ and Co^2+^ sources were added to the reaction solution at the same time, the prepared NiCo-S combined the advantages of Ni-S/NF and Co-S/NF morphology (Figure 4g–i), and the nanosheets were much thinner than Co-S/NF (the nanosheet thicknesses of NiCo-S and Co-S were 10.79 and 144.01 nm, respectively). The NiCo-S structure was staggered in nanoleaves and nanosheets, and the nanosheet structure grew firmly, densely, and homogeneously on the NF. The high-magnification SEM image in Figure 4i shows that the interconnected NiCo-S nanohybrid structure of nanoleaves and nanosheets with an ultrathin thickness of 10.79 nm was vertically grown on the NF substrate, forming a highly porous architecture. This unique structure increased the specific surface area and active site of the material, thus shortening the efficient transfer of electrons at the electrode/electrolyte interface. The interconnected structures could also facilitate the penetration of electrolyte ions onto the crystal surface, which could provide more Faraday reactions on the resulting nanosheets [1]. The distribution of Ni, Co, and S elements in NiCo-S is shown in the EDS mapping image in Figure 4k–m, verifying their presence. Co had a diverging distribution, whereas Ni and S had a comparable distribution. This finding could be attributed to the nanoleaf structures primarily composed of Ni complexes, proving that nanohybrid structures with nanosheets and nanoleaves exist.

The microstructure of Ni-S, Co-S, and NiCo-S was further investigated by analyzing individual nanosheets via TEM, high-resolution TEM (HRTEM), and selected area electron diffraction (SAED). The image of the Ni_3_S_2_ material for TEM (Figure 5a) showed an irregular ultrathin nanosheet structure, indicating that the loose cone loading in SEM was caused by the stacking of irregular nanosheets. In addition, the HRTEM image (Figure 5b) showed that the lattice fringes with a lattice distance of 0.238 nm corresponded to the (003) crystal plane of Ni_3_S_2_. Meanwhile, the SAED image shown in Figure 5c, which was indexed to the (110) and (122) crystal planes of Ni_3_S_2_ in accordance with the diffraction rings, also suggested that the Ni_3_S_2_ substance was polycrystalline. The TEM image of the Co_3_S_4_ material (Figure 5d) showed the nanosheet structure under high magnification, which corresponded to the SEM image. The HRTEM image of the substance (Figure 5e) revealed that the lattice fringes with a lattice distance of 0.285 nm are in agreement with the Co_3_S_4_ (311) crystal plane. The diffraction ring on SAED (Figure 5f) could be indexed to the (220), (311), (400), and (511) crystal planes of Co_3_S_4_. The image of a single NiCo_2_S_4_ nanosheet shown in Figure 5g demonstrated a transparent leaf vein-like morphology, indicating the ultra-thin nature of the material, which is in line with the SEM findings. The dark stripes denoted the foldable edges or nanosheet folds. The image of NiCo_2_S_4_ for HRTEM in Figure 5h showed distinct lattice fringes with lattice distances of 0.235 and 0.216 nm, corresponding to the (400) and (311) crystal planes of NiCo_2_S_4_ (JCPDS: 73-1704), respectively. Meanwhile, the matching SAED pattern depicted in Figure 5i, which features diffraction rings of various radii corresponding to the (311), (400), and (511) crystal planes of NiCo_2_S_4_, indicated the polycrystalline nature of NiCo_2_S_4_.

### 3.2. Electrochemical Performances of NiCo-S/NF Electrode

The electrochemical properties of the prepared Ni-S, Co-S, and NiCo-S were tested in a three-electrode system by using a 6 M KOH solution as the electrolyte, and the results are displayed in Figure 6a. The CV curves of Ni-S, Co-S, and NiCo-S were obtained at a scan rate of 10 mV∙s^−1^ under the potential window ranging from −0.2 V to 0.5 V. A pair of redox peaks could be easily observed for all three CV curves samples, consistent with the Faraday redox reaction mechanism in the process of electrochemical reactions for the reversible intercalation and deintercalation of OH^−^ ions [17,40].

For the N-S/NF electrode material, the redox peaks were located at 0.25 and 0.11 V, respectively, and the related redox reaction may be explained as follows [41,42]:(7)$${Ni}_{3}S_{2}+2{OH}^{-}↔{Ni}_{3}S_{2}{\left(OH\right)}_{2}+2e^{-}$$

For the Co-S/NF electrode material, the redox peaks were located at 0.27 and 0.10 V, respectively, and the corresponding redox reactions is as follows [43]:(8)$${Co}_{3}S_{4}+{OH}^{-}↔{Co}_{3}S_{4}OH+e^{-}$$
(9)$${Co}_{3}S_{4}OH+{OH}^{-}↔{Co}_{3}S_{4}O+H_{2}O+e^{-}$$

Referring to the NiCo-S/NF electrode material, the CV curve showed that the redox peak shifted compared with the two mono-metal sulfides, the area of the CV curve was enlarged, the reduction peak moved to a more negative value (0.09 V), and the oxidation peak remained at 0.27 V. The shift of the redox peak was mainly due to the OH^−^ ion in the electrolyte promoting the Faraday reaction of the redox pair (Ni^2+^/Ni^3+^ and Co^2+^/Co^3+^) in the NiCo-S/NF electrode. The redox reaction corresponded to the following [17,44]:(10)$$Ni{Co}_{2}S_{4}+{OH}^{-}+H_{2}O↔NiS_{4-4x}OH+2CoS_{2x}OH+e^{-}$$
(11)$$CoS_{2x}OH+{OH}^{-}↔CoS_{2x}O+H_{2}O+e^{-}$$

The SC of an electrode material is closely related to the integral area of the CV curve. NiCo-S/NF had a higher integrated area of the CV curve, indicating that the NiCo-S/NF electrode material had a higher SC.

Figure 6b displays the CV curves of NiCo-S/NF at different scan rates from 5 mV·s^−1^ to 50 mV·s^−1^ to further investigate the NiCo-S/NF electrode material. With the increase in scanning rate, the intensity and current density of the CV curve peak also increased, and the current density of the redox peaks corresponded to each other, demonstrating a good Faraday reversible reaction. Meanwhile, as the scanning rates increased, the anode peak shifted in a more positive direction, whereas the cathode peak shifted in a more negative direction. However, the shape of the CV curve essentially remained the same, indicating good rate performance.

In Figure 7a, an obvious potential platform could be observed at the GCD curves of three electrodes measured in 6 M KOH at the current density of 1 A∙g^−1^ within a potential window of 0–0.37 V, confirming the Faraday reaction mechanism. The NiCo-S/NF electrode exhibited longer discharge periods than the other two electrodes. In particular, NiCo-S/NF exhibited a specific capacity of 2081 F∙g^−1^ at 1 A∙g^−1^, which is higher than the specific capacities of Ni-S and Co-S (1520 and 1427 F∙g^−1^, respectively). This finding showed that the NiCo-S/NF electrode exhibited excellent specific capacity. On the one hand, the bimetallic sulfide reduces the energy band gap of the electrode material by using the synergistic effect between metals, which encourages the transfer of charges. On the other hand, the electrochemical performance is affected by the morphology. The SEM diagram showed that the NiCo-S/NF derived from bimetallic NiCo-MOF/NF had a more uniform nanosheet array, and the gap between the nanosheets increased, providing a larger specific surface area. High specific surface area provides higher active sites for Faraday redox reaction. In addition, the use of MOF as a sacrificial template could provide higher porosity to the electrode material, which could shorten the charge transmission distance so that the OH^−^ ions in the electrolyte could easily interpenetrate in the electrode material and promote the participation of NiCo-S/NF in the Faraday redox reaction. When “honeycomb” NF is used as the collector, the higher specific surface area of NF alone could increase the contact area with the active material and improve the utilization rate of the active material, and the higher porosity could shorten the ion transmission distance, thus improving the rate performance of the electrode material.

The constant current charge/discharge curves of NiCo-S at different current densities were measured, as shown in Figure 7b. The SC of the electrode material at different current densities of 1–10 A∙g^−1^ could be calculated using Formula (1). The good rate capability was verified by the specific capacity retention of 58.4%. The SC decreased as the current density was increased because the electrochemical reaction of the sample is affected by the diffusion control. Under low current density, the OH^−^ ions in the electrolyte have sufficient time to participate in the ion intercalation and deintercalation reactions when they interpenetrate the pores of the electrode material. However, at a higher current density, the resulting capacitance loss is mainly due to the increased resistance of OH^−^ ions in the diffusion process, which makes OH^−^ ions unable to diffuse into the pores to the electrode surface in time. Thus, the NiCo-S/NF electrode materials exhibited obvious pseudocapacitance characteristics. On all curves between 0.15 and 0.37 V, a platform could also be seen, suggesting the presence of redox reaction, and further illustrating the major contribution of pseudocapacitance.

The change in SC of the three electrode materials with an increase in current density is shown in Figure 7c. With regard to current densities of 1, 2, 4, 6, 8, and 10 A∙g^−1^, the SCs of Ni-S/NF were 1520.8, 1372.9, 1240, 1148, 1061, and 1000 F∙g^−1^, respectively; those of Co-S/NF were 1427, 1367, 1269, 1200, 1156, and 1124 F∙g^−1^, respectively; and those of NiCo-S/NF were 2081, 1823, 1566, 1407, 1297, and 1216 F∙g^−1^, respectively. The calculation clearly demonstrated that the SC decreased as the current density increased. This phenomenon may be attributed to the insufficient oxidation–reduction process for electrode materials at high current densities, resulting in a rise in resistance. A notable detail is that NiCo-S/NF had a significantly higher SC than Ni-S/NF and Co-S/NF for the same current density. This difference is the result of the synergistic effect of Ni and Co, which increases the proportion of electrochemically active sites and improves the ability of electrode materials to store energy. Meanwhile, the SC (2081 F∙g^−1^ at 1 A∙g^−1^) reported in this research was superior to that of similar electrode materials in the literature (Table 1), indicating that this material has considerable future potential for energy storage.

EIS measurements were performed to explore the conductivity of the electrodes. Figure 7d shows the Nyquist plots of the three samples and the equivalent circuit illustration of the NiCo-S/NF electrode. EIS was measured in a frequency ranging from 100 kHz to 0.001 Hz. The impedance (EIS) plots of the three electrodes were compared in the same plot to visually verify the charge transfer and electrolyte diffusion process at the electrode/electrolyte interface (inset is the fitted equivalent circuit diagram). The curves of the three electrode materials were composed of a straight line and a semicircle. Equivalent series resistance (ESR), also known as Rs, is the semicircle’s intersection with the real axis (Z′), and it reflects the impedance of the electrode material, electrolyte resistance, and electrode/collector contact resistance [49]. The semicircle diameter at the high-frequency region is the charge transfer resistance (Rct), which represents the electron transfer kinetics of the electrode material when it comes into contact with the electrolyte solution. The diagonal line is in the low-frequency region, representing the Warburg resistance (Wo), which is described as the diffusion rate of active ions within the electrode materials. The impedance values of NiCo-S, Ni-S, and Co-S are shown in Table 2. The NiCo-S electrode exhibited lower intrinsic resistance (0.41 Ω) and charge transfer resistance (0.13 Ω) than the Ni-S and Co-S electrodes, indicating that NiCo-S/NF could effectively reduce the contact resistance and increase the charge migration. In addition, NiCo-S/NF exhibited a higher slope of the line in the low-frequency region, where a higher slope is associated with a smaller resistance to ion diffusion and a lower Warburg impedance; the nanosheet structure of NiCo-S was more uniform, and the hybrid structure of nanosheet@nanoleaf could provide a larger specific surface area and more pore structure, which make ion diffusion easier to carry out.

The electrochemical stability of electrode materials is also one of the key parameters to evaluate the performance of supercapacitors. The Coulomb efficiency of NiCo-S/NF remained at 97.9%, indicating that the electrode material had a highly reversible Faraday redox reaction [50]. The cycling experiment under 50 mV∙s^−1^ at the voltage window ranging from −0.2 to 0.6 V further displayed that NiCo-S/NF presented good electrochemical stability, as shown in Figure 8. The attenuation was around 23% in the first 3000 cycles. The main reason is that during the prolonged redox reaction process, the electrode material continuously contracts and expands, which causes a little exfoliation of the active material from the collector [51]. Simultaneously, throughout the lengthy cycle, some nanoarrays partially collapsed, which slows down OH^−^ diffusion and lowers the specific capacity [52]. The NiCo-S/NF SC remained at 75.7% of the initial SC after 5000 cycles. Compared with those in other studies in the literature (Table 3), the NiCo-S/NF electrode demonstrated good cycling stability. The SEM image of the electrode material after 5000 cycles showed that the NiCo-S nanosheet array nearly retained its original morphology. The SEM image of NiCo-S/NF after 5000 cycles was shown in the inset of Figure 8. Compared with Figure 4g,i, the electrode material falls off from NF and the nanosheet roughness increases after cycling, which further explains the stable reduction of cycling of the electrode material.

### 3.3. Electrochemical Measurements of NiCo-S//AC HSC Device

The above results indicated that NiCo-S/NF showed excellent electrochemical performance. A NiCo-S/NF//AC HSC device was constructed to further display the real application of the NiCo-S/NF material. Figure 9 is the simplified scheme of the HSC device in 6 M KOH, in which the synthesized NiCo-S/NF//AC material was used as the positive electrode. Meanwhile, the AC/NF electrode served as the negative electrode owing to its high specific surface and excellent electrical conductivity. A filter paper was used as a diaphragm in the middle. In the process of preparing the two-electrode device, the required AC mass (excluding the NF load) was calculated using Formulas (3) and (4) so that the two-electrode device could exhibit excellent performance. A reasonable mass ratio of positive and negative active materials could ensure compliance with the law of charge conservation in the electrochemical reaction process and further enhance the performance of the materials.

The prepared negative electrode (AC/NF) was assessed for its electrochemical activity by using a two-electrode system with 6 M KOH solutions as the electrolyte. Figure 10a shows that the AC/NF electrode possessed a rectangular shape at a scan rate of 50 mV·s^−1^, demonstrating a usual electric double-layer capacitance property. The specific capacity of AC, as calculated by Formula (3), was 108.8 F∙g^−1^.

In Figure 10b, the CV curves of NiCo-S/NF and AC/NF electrodes were examined at a scan rate of 50 mV·s^−1^ to verify the voltage window of the NiCo-S/NF//AC HSC assembly. The voltage window of AC/NF was from −1.0 V to 0 V, and the voltage window of NiCo-S was from −0.2 V to 0.5 V, indicating that the potential stability of the NiCo-S/NF//AC HSC voltage window may be 1.7 V. The CV and GCD curves of HSC were measured to further prove the voltage window of NiCo-S/NF//AC HSC. The electrical performance of NiCo–S//AC HSC was evaluated at various voltage ranges of 0–1.3 and 0–1.7 V with a scan rate of 50 mV·s^−1^, as seen in Figure 10c. The CV curve at 1.7 V differed from the curves of 1.3–1.6 V, which could be attributed to the polarization of the electrode material, demonstrating that the voltage window of 1.6 V was suitable for the NiCo-S/NF//AC HSC device. As the voltage increased to 1.7 V, the GCD curves (Figure 10d) indicated that side reactions may be present as the plateau feature of the GCD curve became more prominent. Therefore, the constructed HSC device was stable at a voltage of 1.6 V. These findings ascertained that 1.6 V was the most effective voltage to utilize for further electrochemical study. Figure 10e presents the CV curves of NiCo-S/NF//AC HSC from 5 mV·s^−1^ to 50 mV·s^−1^ with a voltage of 1.6 V. The CV curves at various scan rates still showed redox peaks, indicating the battery-like behavior of the NiCo-S/NF//AC HSC device. Furthermore, the graphs showed no noticeable changes when the scan rate was increased from 5 mV·s^−1^ to 50 mV·s^−1^, indicating that the assembled HSC had excellent electrochemical reversibility. The HSC was also examined by a GCD test at various current densities ranging from 0.5 A·g^−1^ to 10 A·g^−1^ (Figure 10f). The results revealed that the assembled HSC could reach a voltage of 1.6 V, suggesting its potential use in actual supercapacitor implementations. Furthermore, no distinct platform could be seen on the GCD curves, and the non-linearity of the curve further indicated that the results were influenced by the combination of faradaic redox capacitance and electrochemical double-layer capacitance [40]. As calculated by Formula (4), the SCs of the HSC device were 60.3, 55.5, 48.8, 40.2, 33.4, and 28.5 F·g^−1^ at the current densities of 0.5, 1, 2, 4, 6, and 8 A·g^−1^, respectively. The SC of the HSC device could maintain 45.74% even at a current density of 8 A·g^−1^, conforming to its excellent rating performance.

Figure 11a displays the Coulomb efficiency of the HSC device under long cycles. After 3000 cycles, the Coulomb efficiency was 98.8%, demonstrating excellent reversibility. For evaluation of the overall performance of the NiCo-S/NF//AC asymmetric supercapacitor, the energy density and power density of the HSC device were drawn from the Ragone plots in Figure 11b. In accordance with Formulas (5) and (6), Figure 11b displays the highest energy density of 25.58 Wh kg^−1^ at a power density of 400 W kg^−1^ and at a current density of 0.5 A·g^−1^, and the lowest energy density of 11.2 W kg^−1^ at a power density of 6400 W kg^−1^ when the current density was increased to 8 A g^−1^. The performance of the device developed here was much larger than the values in other previous reports, such as NiCoS/NF//AC (11.6 Wh∙kg^−1^ at 3762 W∙kg^−1^) [55], Ni_3_S_2_@CoS//AC (14.4 Wh∙kg^−1^ at 1075.52 W∙kg^−1^) [56], NiCo_2_S_4_/HMCSs//HMCSs (13.2 Wh∙kg^−1^ at 3714 W∙kg^−1^) [57], CoNi_2_S_4_/Ni/OTL//CM/Ni/OTL (10.6 Wh∙kg^−1^ at 2488.3 W∙kg^−1^) [58], and NiCo_2_S_4_/GA//AC (20.9 Wh∙kg^−1^ at 800.2 W∙kg^−1^) [59]. These encouraging electrochemical performances of NiCo-S/NF//AC HSC herein showed its potential application in supercapacitors and energy storage devices.

## 4. Conclusions

In conclusion, a novel nanosheet@nanoleave hybrid of bimetallic sulfide was successfully designed and synthesized by using NiCo-MOF as a self-sacrificing template. The results show that compared with single-metal sulfide, bimetallic sulfide greatly improves the specific capacity of electrode materials and has good magnification performance due to the synergistic effect between the two metals and the rich redox reaction of electrode materials. After 5000 cycles at the scan rate of 50 mV·s^−1^, the capacity retention rate was 75.7%. In addition, the assembled NiCo-S/NF//AC HSC device can operate at a voltage window of 1.6 V and an energy density of up to 22.58 Wh·kg^−1^ at a power density of 400 W·kg^−1^. The results show that NiCo-S/NF electrode materials have great application potential in supercapacitors.

## Figures and Tables

**Figure 1 nanomaterials-13-01229-f001:**
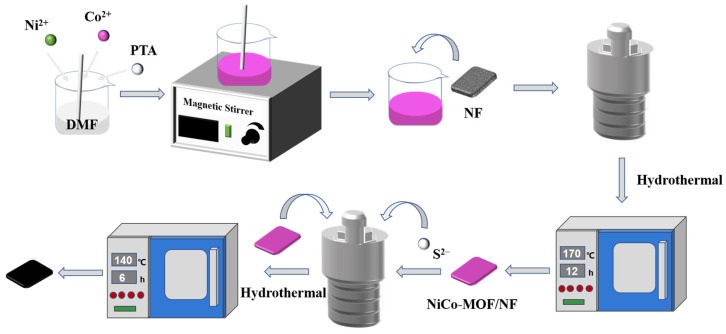
Illustration of the preparation process from NiCo-MOF to NiCo-S.

**Figure 2 nanomaterials-13-01229-f002:**
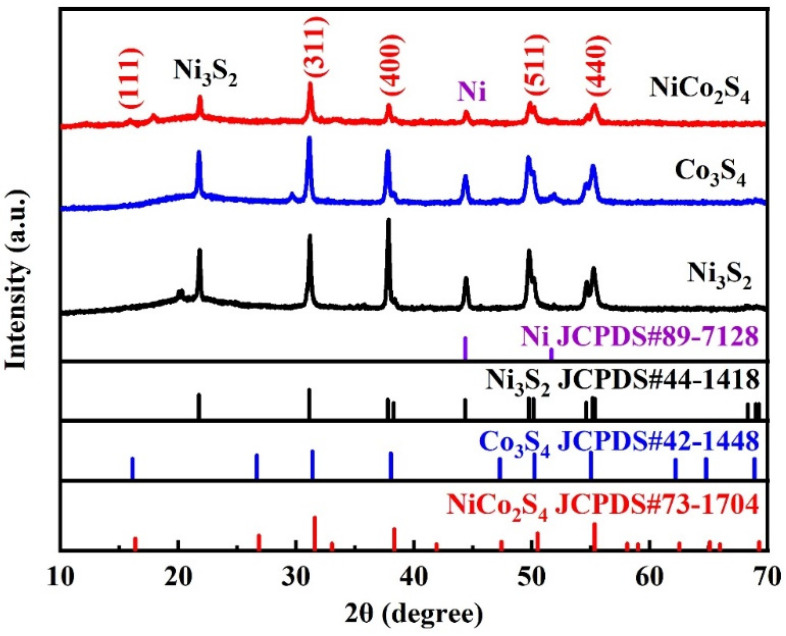
XRD patterns of samples.

**Figure 3 nanomaterials-13-01229-f003:**
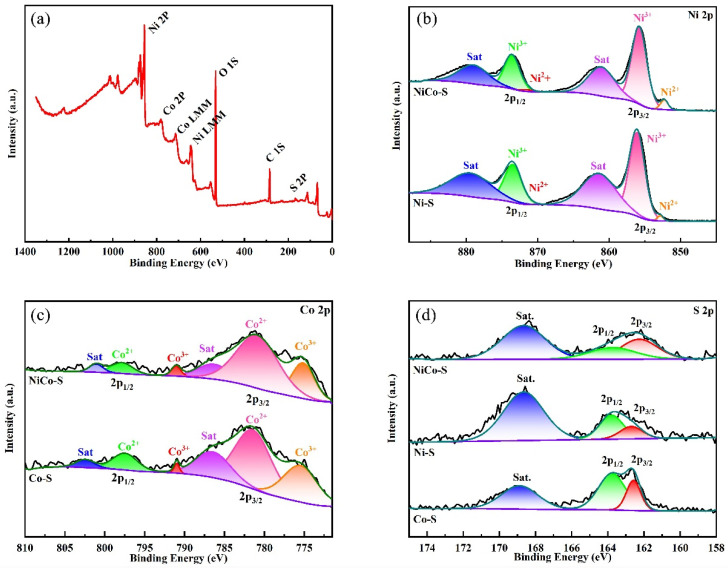
(**a**) XPS survey spectra of NiCo-S/NF electrode, high-resolution XPS spectra of NiCo-S/NF, Ni-S/NF, and Co-S/NF of (**b**) Ni 2p, (**c**) Co 2p, and (**d**) S 2p.

**Figure 4 nanomaterials-13-01229-f004:**
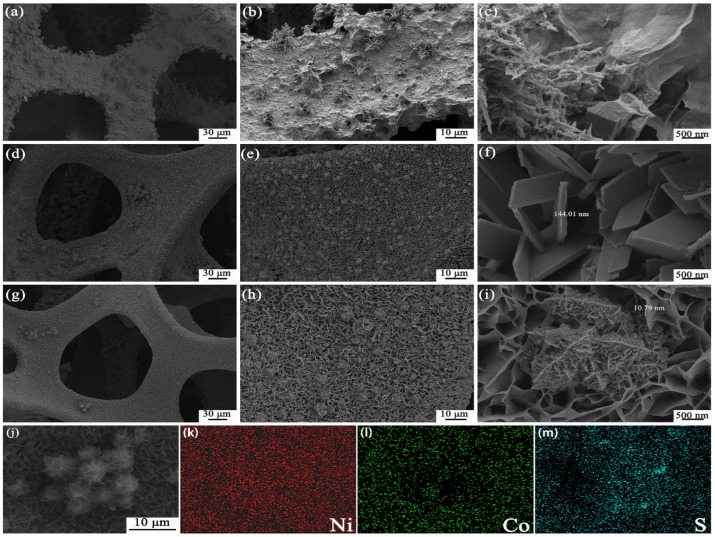
SEM images of the prepared samples: (**a**–**c**) Ni-S/NF; (**d**–**f**) Co-S/NF; (**g**–**i**) NiCo-S/NF; (**j**) FESEM image and the corresponding EDS elemental mappings for (**k**) Ni, (**l**) Co, and (**m**) S.

**Figure 5 nanomaterials-13-01229-f005:**
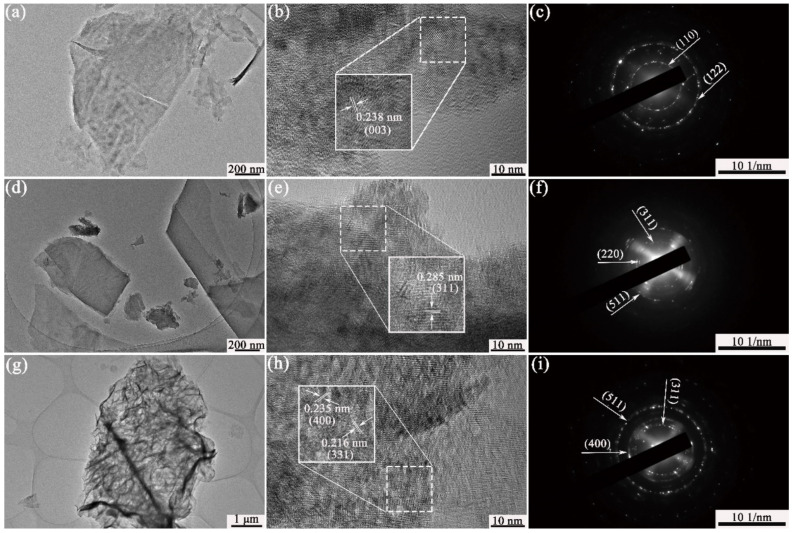
TEM, HRTEM, and SAED of the prepared samples: (**a**–**c**) Ni-S/NF; (**d**–**f**) Co-S/NF; (**g**–**i**) NiCo-S/NF.

**Figure 6 nanomaterials-13-01229-f006:**
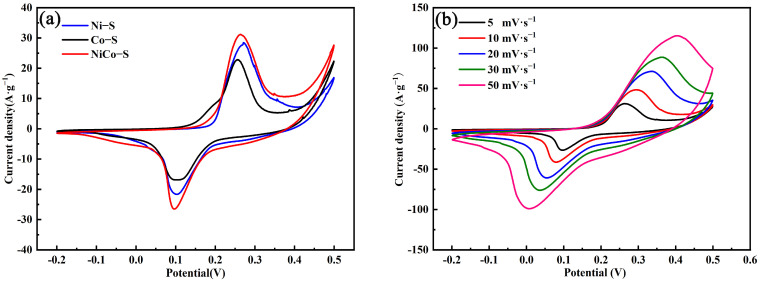
(**a**) CV curves of the prepared samples for Ni-S, Co-S, and NiCo-S at a scan rate of 10 mV∙s^−1^. (**b**) CV curves at different scan rates of NiCo-S.

**Figure 7 nanomaterials-13-01229-f007:**
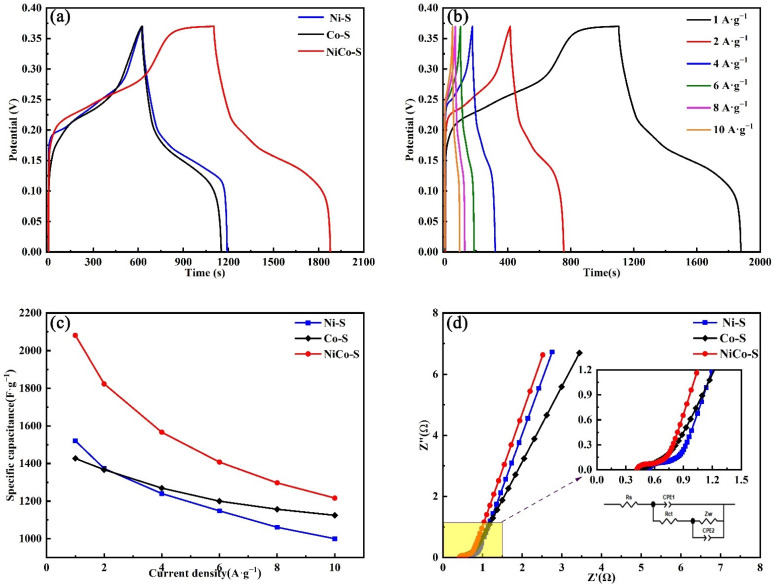
(**a**) GCD curves at 1 A∙g^−1^ of different electrodes. (**b**) GCD curves at different current densities of the NiCo-S electrode. (**c**) Variation in specific capacitance of all electrodes with different current densities. (**d**) EIS of all electrodes.

**Figure 8 nanomaterials-13-01229-f008:**
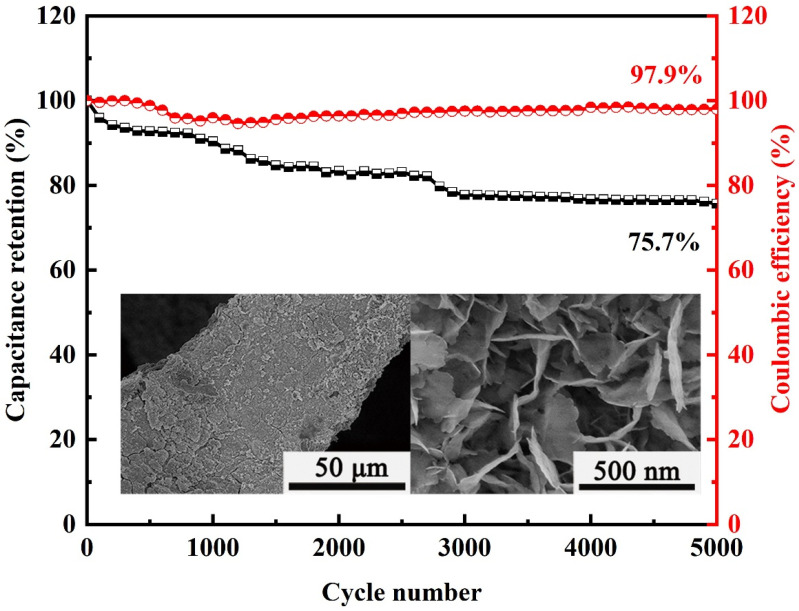
Cycling stability for NiCo-S/NF at the scanning rate of 50 mV·s^−1^ up to 5000 cycles.

**Figure 9 nanomaterials-13-01229-f009:**
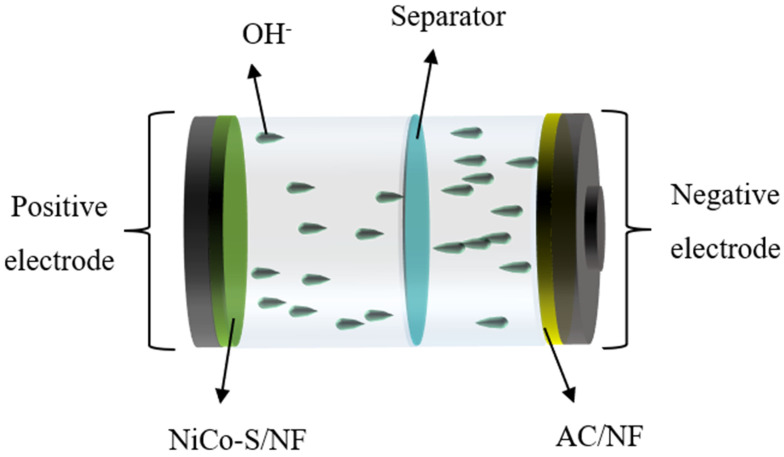
Schematic illustration of the assembled hybrid supercapacitor device.

**Figure 10 nanomaterials-13-01229-f010:**
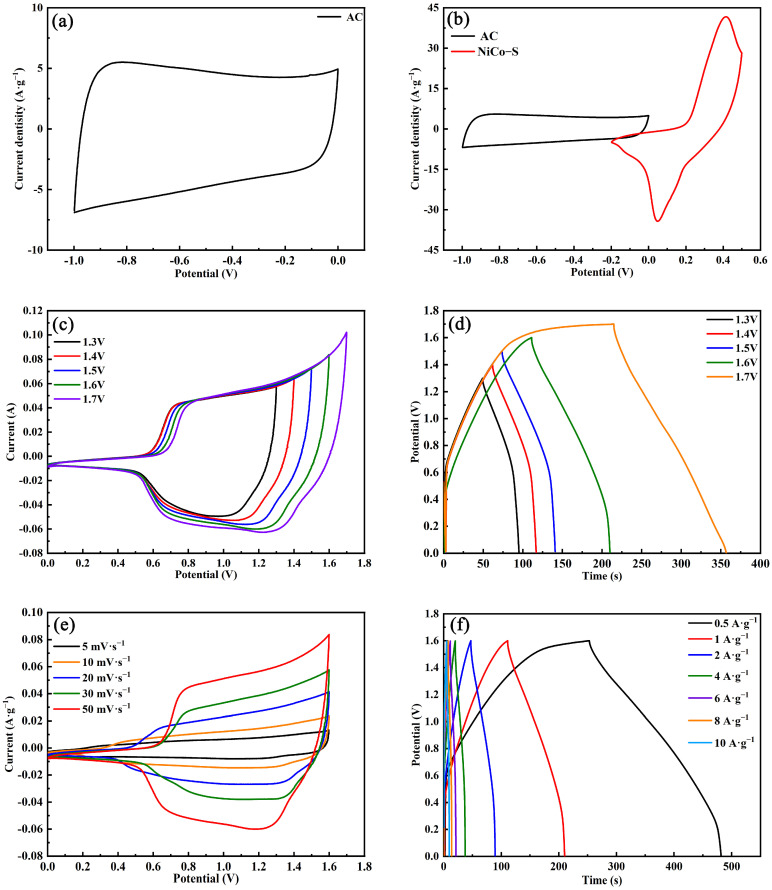
(**a**) CV curves of AC in a three-electrode system at a scan rate of 50 mV·s^−1^. (**b**) CV curves of NiCo-S and AC electrodes at a scan rate of 50 mV·s^−1^. (**c**) CV curves of HSC device at 50 mV·s^−1^ with different potential windows. (**d**) GCD curves of HSC device at 1 A·g^−1^ with different potential windows. (**e**) CV curves of HSC devices with various scan rates. (**f**) GCD curves of HSC device at different current densities.

**Figure 11 nanomaterials-13-01229-f011:**
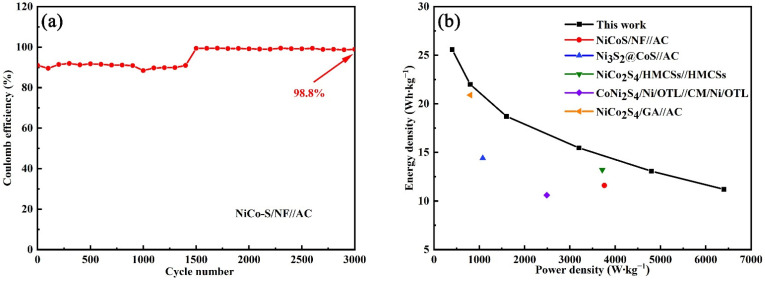
(**a**) Coulomb efficiency of NiCo-S/NF//AC HSC at a scan rate of 50 mV·s^−1^ after 3000 cycles. (**b**) Ragone plot of assembled NiCo-S/NF//AC HSC to compare the performances of other supercapacitors.

**Table 1 nanomaterials-13-01229-t001:** The properties of NiCo-S/NF are compared with those reported in other studies in the literature.

Electrode	Morphology	Electrolyte	Performance	Refs.
NiCo-S/NF	Nanosheet@nanoleaves	6 M KOH	2081 F∙g^−1^ 1 A∙g^−1^	This work
NiCo_2_S_4_	Nanoflowers	6 M KOH	1757 F∙g^−1^ 1 A∙g^−1^	[45]
NiCo_2_S_4_	Urchin-like	6 M KOH	1149 F∙g^−1^ 1 A∙g^−1^	[46]
CoNi_2_S_4_@CC	Nanowire	6 M KOH	1872 F∙g^−1^ 1 A∙g^−1^	[47]
NiCo_2_S_4_@NC	Nanoflower-like	6 M KOH	1021.6 F∙g^−1^ 1 A∙g^−1^	[48]

**Table 2 nanomaterials-13-01229-t002:** Parameters of the equivalent circuit mod.

Samples	NiCo-S/NF	Ni-S/NF	Co-S/NF
Rs	0.41	0.53	0.44
Rct	0.13	0.14	0.27
Wo	0.53	0.60	0.83

**Table 3 nanomaterials-13-01229-t003:** Comparison of capacity retention rate of electrode materials after long cycle with other studies in the literature.

Materials	Cycle Number	Capacitance Retention	Refs.
NiCo-S/NF	5000	75.7%	This work
NiCo-S/NF	1000	71.8%	[53]
NiCo_2_S_4_@C	3000	71.4%	[27]
NiCo_2_S_4_@PANI	5000	61.64%	[54]
NiCo_2_S_4_/NF	1000	66%	[52]

## Data Availability

Not applicable.

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
