# Peer review of "MOF-Derived Ultrathin NiCo-S Nanosheet Hybrid Array Electrodes Prepared on Nickel Foam for High-Performance Supercapacitors"

_nanomaterials, 2023, doi:10.3390/nano13071229_

Round 1

Reviewer 1 Report

Comments to the Author

The authors reported an interesting work about “MOF Derived Ultrathin NiCo-S Nanosheet Hybrid Arrays Electrodes Prepared on Nickel Foam for High-performance Supercapacitors. In all, this study is interesting and meaningful. It can be recommended to be published after revisions.

1.     Authors reported the pseudocapacitive type material. So, the eqn. 1 is wrong to calculate specific capacitance from CV analysis.

2.     The authors should perform the GCD analysis with the potential window of -0.2 to 0.5 V as performed in CV.

3.     Authors should calculate the coulombic efficiency of the device.

4.     Authors should calculate ESR value from the nyquist plot.

Reviewer 2 Report

The authors present an analysis of novel NiCo-S nanostructured hybrid electrode material for supercapacitors, which uses MOFs as a sacrificial template. A nickel foam is used as supporting macroporous material. Good electrochemical performance is achieved compared with previous works. The work is of potential interest to the journal. Revision of the following aspects is requested:

1.    Some writing typos must be corrected. Por example, on page 5, there is no reason to use “so as to obtain” in italics. On page 8, “It is proven that that” must be “It is proven that”.

2.    The authors must further explain the effect of electrolyte concentration on Table 2 when comparing electrochemical performance with previous work.

3.     The evolution of the multiscale porous structure of the electrodes over time during cycling should be discussed to further analyze the observed results. Although agglomeration and clogging of nanoporous structure can be anticipated, it will be convenient to establish a comparison between the prepared electrodes. Currently, only cycling of one electrode is examined.

4.     The authors must provide the specifications of the Ni foam used as supporting porous material (porosity and specific surface area) and discuss the effect of the supporting layer on performance if it has been already examined as part of their work.
